∂ | **Open Peer Review** | Bacteriology | Research Article

# Nontuberculous mycobacteria remodel lung microbiota in cystic fibrosis-associated respiratory infections

Michelle Hardman,[1] Sarah Higgi,[2] Liam Hanson,[1] Kristin Schutz,[3] Matthew J. Wargo,[3] Charlotte C. Teneback,[4] Thomas W. V. Daniels,[5,6] Christopher van der Gast,[7,8] Damian W. Rivett[9]

**ABSTRACT** Nontuberculous mycobacterial (NTM) infections in people with cystic fibrosis (pwCF) can have detrimental effects on prognosis and pose significant challenges to treatment. However, there are still questions regarding the contribution and influence of NTMs on the respiratory microbiome and the mechanisms by which NTMs cause infections. Here, we investigate the impact of NTM infection on microbiome composition and lung function (percent predicted forced expiratory volume in 1 second). Primary comparisons were between culture-positive cohorts for *Mycobacterium avium* complex and *Mycobacterium abscessus* complex and those who were culture-negative for NTMs and attending outpatient clinics. Additionally, the consequence of cystic fibrosis transmembrane conductance regulator (CFTR) modulator therapy status and positive NTM culture was assessed in terms of microbiome change. Our data suggest that the presence of NTM significantly alters the diversity and the composition of the lung microbiota in pwCF, including those receiving CFTR modulator therapies. Importantly, significant associations were detected between NTM presence and changes in abundance of *Pseudomonas aeruginosa* and *Burkholderia cepacia* complex members, inferring modulatory effects of NTMs on respiratory microbiomes. This study contributes to the understanding of NTM infection and these organisms' interaction with the respiratory microbiome and CFTR modulator therapy, highlighting the need for further research in this area.

**IMPORTANCE** The influence of NTM infection in pwCF is still debated, and the extent of their contribution to mortality and morbidity is still questioned. Findings in this study highlight a link between the presence of NTMs and significant alterations in the composition of the respiratory microbiome, particularly with respect to some of the canonical CF pathogens, especially *Pseudomonas aeruginosa* and members of the *Burkholderia cepacia* complex. This indicates that complex relationships are occurring within the microbiome. This study further contributes to the understanding of NTM infection in pwCF, with and without CFTR modulator therapy, and highlights the need for further research in this area. The knowledge gained from this study has implications for treatment strategies and management, ultimately aiming to improve and prolong the lives of pwCF.

**KEYWORDS** Nontuberculous mycobacterium, cystic fibrosis, *Pseudomonas aeruginosa*, *Burkholderia cepacia* complex, CFTR modulator therapy, microbiome, respiratory infection

Address correspondence to Damian W. Rivett, d.rivett@mmu.ac.uk, or Christopher van der Gast, chris.vandergast@northumbria.ac.uk.

Outside of this published work DWR and CvdG have received funding from Vertex Pharmaceuticals Ltd. The rest of the authors declare no competing interests.

See the funding table on p. 13.

Cystic fibrosis (CF) is a multisystemic genetic disorder affecting more than 70,000 people worldwide (1). This autosomal recessive disease is caused by mutations in the cystic fibrosis transmembrane conductance regulator (CFTR) gene (2) leading to an accumulation of abnormally viscous mucus in several major organ systems. The result

of this is a variety of symptoms affecting the whole body. People with CF typically have several pulmonary symptoms such as recurrent chest infections, coughing, trouble breathing, and wheezing (1), which contribute to respiratory disease as the primary cause of morbidity and mortality (3). While CFTR modulators are relatively new to the CF treatment regimen, they can enhance the expression, function, and stability of a faulty CFTR protein (4). However, bacterial infection remains a constant issue (5), and research has not yet fully elucidated the effect that modulator therapy has on the respiratory microbiome.

Nontuberculous mycobacteria (NTMs) are ubiquitous environmental organisms that can cause chronic pulmonary infection in people with cystic fibrosis (pwCF). Once infected with NTMs, pwCF are more likely to develop severe lung disease and experience complications than those in the general population (6–8); however, those colonized by NTMs do not always have active disease (9). Where colonization progresses into active disease, pwCF have shown a significant reduction in percentage predicted of forced expiratory volume in 1 second (%FEV$_1$) and increased frequency of exacerbations (10, 11) and may be ineligible for lung transplantation due to the intrinsic antimicrobial properties of some NTM species (12).

The severity of nontuberculous mycobacterial pulmonary disease (NTM-PD) is highly dependent on the type of NTM acquired. One of the most clinically relevant, rapidly growing mycobacteria is the *Mycobacterium abscessus* complex (MABSC). The detection and isolation of MABSC has been increasing globally (13), as it is associated with increasing morbidity and mortality rates in immunocompromised individuals and those with underlying pulmonary diseases (14, 15). Conversely, the *Mycobacterium avium* complex (MAC) is part of the slow-growing mycobacterial group often isolated from soil, water, birds, and livestock (16). MAC infection often exhibits less aggressive disease and better pwCF outcomes when compared to MABSC (17). Therefore, the accurate and timely diagnosis of the type of NTM infection is essential to manage disease and prevent further damage to the pulmonary system (9, 18).

MAC and MABSC are associated with around 90% of the total reported cases of NTM-PD (19–21). The recent estimated global prevalence of NTM infection in pwCF is approximately 7.9%, with MABSC infection estimated at 4.1% and MAC at 3.7% (22). In 2018, NTM prevalence was increasing by 5% annually in the US CF population, driven mainly by MAC infection (23) and with a 2.5% rise over a 5-year period in the UK (24), with MABSC being the predominant species detected (25). NTM-PD is the most common type of NTM infection globally and accounts for 80%–90% of all NTM-associated diseases (26–29).

The presence of NTMs and their association with other CF pathogens and the diversity of the CF microbiome have not been a major research focus, despite evidence that lung infection in CF is unquestionably polymicrobial in nature (30–34). Previous studies examining the interplay between NTM populations and NTM-PD in CF micro-biomes are sparse; there is, however, limited research into NTM-microbiome associations in other pulmonary disorders that can contextualize this work. Macovei et al. (35) found that NTMs, including opportunistic pathogens, were present in healthy participants and that Streptococcaceae and Staphylococcaceae constituted a significant proportion of the microbiota. Yamasaki et al. (36) discovered that pwCF positive for NTM had a microbiota predominantly composed of *Prevotella*, *Streptococcus*, *Neisseria*, and *Pseudomonas*, and that the incidence of anaerobes was higher in those diagnosed with NTM infection. This suggests that anaerobes may play a role in the pathogenesis of NTM disease.

While there have been other studies examining the composition of the microbiota in the presence of NTMs, with most suggesting a unique bacterial community residing within each pwCF (35–37) or the impact of CFTR modulator therapies on NTM prevalence (38), there has been no research combining NTM complexes, CFTR modulator therapies, and CF-associated lung microbiome.

Here, we investigated changes in the CF lung microbiome during NTM infection and CFTR modulator therapy. Using a combination of clinical, diagnostic microbiology,

and microbiota sequencing data, we demonstrate the remodeling of the microbiome undertaken both by NTM infection and CFTR modulator therapy, with the significant reduction of some key pathogens (*Pseudomonas aeruginosa*) and the emergence of others (*Burkholderia cepacia* complex). This knowledge will enhance the understanding of how NTMs influence other pathogens, providing information regarding CF lung disease progression in relation to the microbiome in the presence of CFTR modulator therapy.

## RESULTS

Due to the complexity of the sample isolation of pwCF during the coronavirus disease 2019 (COVID-19) pandemic, it was necessary to combine results from sputum samples and cough swabs, some of which were collected in the clinic, while others were mailed (Table 1). The impact of this mixed sampling approach was therefore tested. There were no significant differences between the lung function (measured as %FEV$_1$) of pwCF and sample type ($F_{1,47} = 0.09$, $P = 0.768$) or collection method ($F_{1,47} = 0.01$, $P = 0.912$). Our analysis, therefore, focused on the influence of clinical characteristics and the impact of chronic NTM-positive culture on lung function from a cohort of 57 pwCF taken from the UK and the USA (Table 1). Here, we found no statistically significant difference between the lung function of pwCF and age at sampling ($F_{1,47} = 0.00$, $P = 1.000$) or whether sampling occurred during an exacerbation ($F_{1,47} = 0.74$, $P = 0.395$), location ($F_{1,47} = 1.61$, $P = 0.211$), sex ($F_{1,47} = 0.07$, $P = 0.788$), whether the pwCF was being treated with modulator therapy ($F_{1,47} = 1.15$, $P = 0.290$) or antibiotics ($F_{1,47} = 2.74$, $P = 0.105$) at the time of sampling, or whether the sputum was positive for NTM culture ($F_{1,47} = 0.70$, $P = 0.408$). When the analysis considered an interactive effect of NTM infection and modulator therapy, a higher lung function was recorded for the group without NTM infection undergoing CFTR modulator therapy (%FEV$_1$ of 73.1 ± 29.6) than any other combination (NTM negative, no modulator, %FEV$_1$ of 55.1 ± 21.7; NTM positive, no modulator, %FEV$_1$ of 65.8 ± 24.3; NTM positive, undergoing modulator therapy, %FEV$_1$ of 58.3 ± 27.1); however, no significant interactions were observed ($F_{3,49} = 1.10$, $P = 0.359$). Furthermore, there were no significant differences when assessing whether changes in

**TABLE 1** Summary of pwCF clinical characteristics[a]

| Characteristics | Number of participants |
| --- | --- |
| Number of pwCF | 57 |
| Southampton, UK (%) | 33 (58) |
| Burlington, VT (%) | 24 (42) |
| Collection method (clinic/posted) | 16/41 |
| Sample type (sputum/cough swab) | 35/22 |
| Sex (male/female) | 18/39 |
| Mean (SD) age (years) | 29.2 (±6.6) |
| Minimum to maximum age (years) | 19–53 |
| Mean %FEV$_1$ [b] (SD) | 64.8 (27) |
| Individuals on CFTR modulator therapy (%) | 30 (53) |
| CFTR genotype | |
| Homozygous Phe508del (%) | 29 (51) |
| Heterozygous Phe508del (%) | 23 (40) |
| Non-Phe508del (%) | 5 (9) |
| Individuals with chronic positive NTM[c] culture (%) | 27 (47) |
| MAC[d] (%[f]) | 14 (52) |
| MABSC[e] (%[f]) | 11 (41) |
| Other (%[f]) | 5 (19) |

[a]Data are presented as mean and standard deviation (SD) or number and percentage (%) unless otherwise stated.
[b]%FEV$_1$, percentage predicted of forced expiratory volume in 1 second.
[c] NTM, Nontuberculous mycobacteria.
[d]MAC – *M. avium* complex.
[e]MABSC – *M. abscessus* complex.
[f]Percentage of chronic NTM culture-positive pwCF.

microbial diversity impacted lung function for either diversity ($R = 0.11$, $P = 0.420$) or dominance ($R = -0.07$, $P = 0.625$).

## Microbiome diversity changes with NTM-positive culture and modulator treatment

The 16S rRNA gene sequencing yielded a total of 415,856 bacterial sequence reads after filtering and quality control (39), with a mean (±standard deviation throughout) number of 7,296 (±6,283) reads per pwCF ($n = 57$, range = 1,180–31,791 reads). All 16S rRNA reads from NTMs were also removed so as not to bias the analysis. In total, 215 bacterial operational taxonomic units (OTUs) were assigned after manual curation (34) with a mean of 12.3 (±8.7) OTUs per sample.

While sample storage has been shown as not having an impact on the dominant members of the microbiome (40, 41), sample type is known to have significant discrepancies in microbiome analysis in adults (42). To acknowledge this, we analyzed the impact of sample type on the microbiome, determining that there was a significant difference in microbial composition in terms of diversity ($t_{26} = 4.09$, $P < 0.001$) and dominance ($t_{54} = 4.68$, $P < 0.001$). However, accounting for this is not trivial. As with other studies (43–45), we found that sputum production is inversely associated with modulator therapy (odds ratio [OR] = 0.04, 95% confidence interval [CI] 0.01–0.17, $P < 0.001$) and associated with exacerbations (OR = 15.00, 95% CI 3.67–103.72, $P = 0.001$). As the objective of this study is to begin to understand the effect of chronic NTM infection on the respiratory microbiota across the pwCF spectrum, we have combined the samples for the analysis so as not to bias the results for a particular populace (45).

Furthermore, the effect of antibiotic treatment (binary) at the time of sampling was assessed. In our data set, antibiotic treatment was more likely to be the case for pwCF producing sputum samples (OR = 5.82, 95% CI 1.87–20.21, $P = 0.003$). The results indicated that antibiotics had a significant effect on diversity ($t_{37} = 2.56$, $P = 0.015$) but not dominance ($t_{55} = 1.68$, $P = 0.099$).

Finally, the likelihood of sputum production being associated with being NTM culture positive was also assessed (OR = 8.50, 95% CI 2.61–32.03, $P = 0.001$); however, there was no significant increase in the likelihood of being on antibiotics if NTM was culture positive (OR = 1.88, 95% CI 0.66–5.49, $P = 0.242$). Due to this and the confounding effects of the other variables, we subsequently accounted for antibiotic treatment in all further models to address the issue of different sampling strategies while retaining numbers to generalize the effect of NTM infection on as broader a range of pwCF as possible. Interestingly, when the NTM culture status was considered, there were significant changes in diversity ($t_{33} = 3.59$, $P = 0.001$) and community dominance ($t_{53} = 2.07$, $P = 0.044$), with the significant effect of NTM-positive culture retained after accounting for antibiotic treatment ($F_{1,54} = 12.06$, $P = 0.001$). The analysis indicated that samples which were found to be culture positive for NTMs (Fig. 1A) had a lower diversity (mean Fisher's alpha index of diversity = 1.04 ± 0.48) while being more dominated by a single taxon (mean Berger-Parker index of dominance = 0.44 ± 0.16) compared to the culture-negative samples (mean Fisher's alpha = 1.95 ± 1.24, mean Berger-Parker index = 0.34 ± 0.21).

Further investigation focused on what impact the group of NTM present had on the microbiome diversity (Fig. 1B). The results showed that, after accounting for significant ($F_{1,52} = 8.22$, $P = 0.006$) antibiotic treatment, the NTM type had significant effects ($F_{3,52} = 4.31$, $P = 0.009$) on the microbiome. Furthermore, pwCF diagnosed with MABSC were found to have significantly ($P_{adj} = 0.010$) lower diversity (mean Fisher's alpha = 0.96 ± 0.31) than the NTM-negative group (mean Fisher's alpha = 1.95 ± 1.24). No significant differences were observed between the NTM-negative group and the MAC group (mean Fisher's alpha = 1.03 ± 0.55, $P_{adj} = 0.103$) and other NTMs cultured (mean Fisher's alpha = 1.24 ± 0.57, $P_{adj} = 0.509$). Furthermore, no significant difference was observed between any of the NTM types ($P_{adj} > 0.751$). The NTM type did not significantly ($F_{3,65} = 1.28$, $P = 0.289$) influence how dominated a community was.

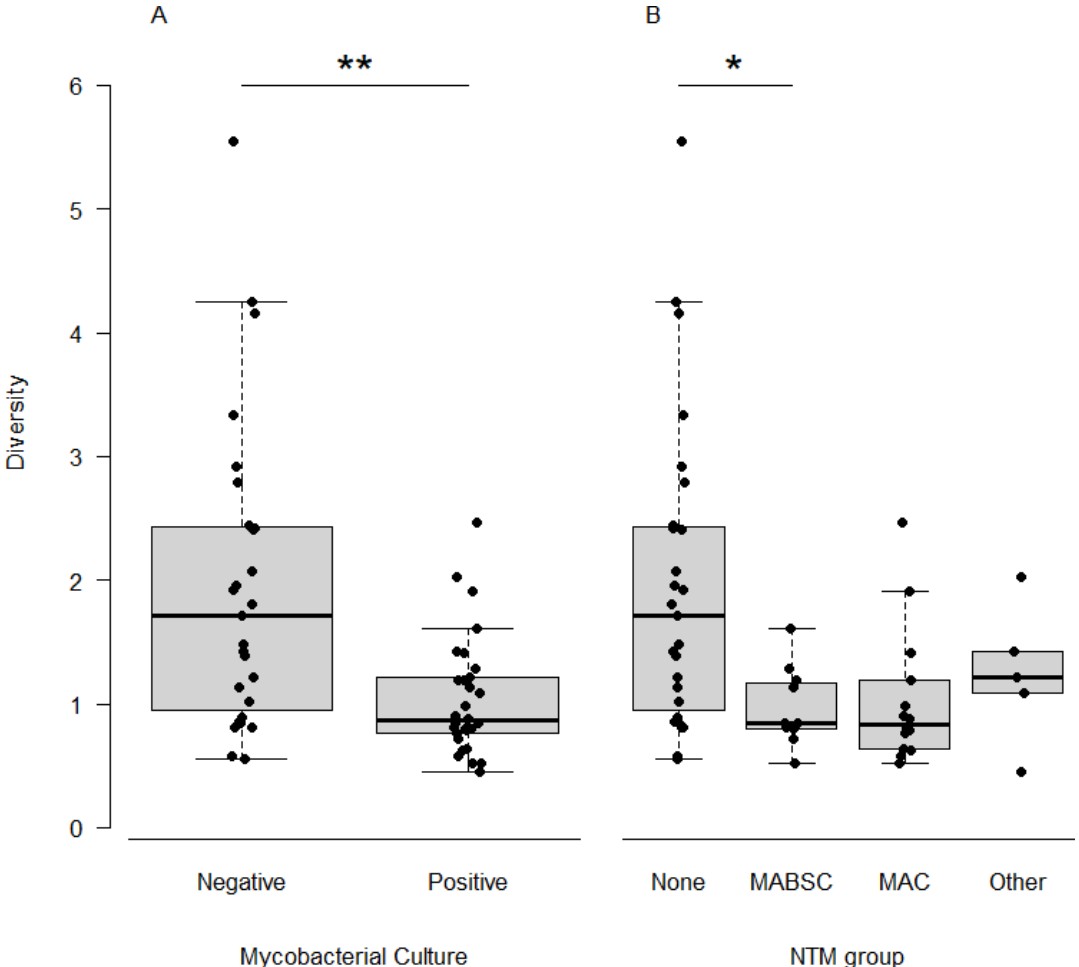

FIG 1 Differences in microbiome diversity during NTM infection. Differences in diversity (Fisher's alpha) were shown to be significant in NTM culture-positive and NTM culture-negative samples (A); however, there was little impact beyond this when the different NTM groups were considered (B). The asterisk indicates a statistically significant Tukey's honestly significant difference result, where * and ** represent $P$ values of <0.05 and 0.01, respectively.

The analysis then turned to the assessment of modulator treatment on the microbiome and whether there was any interplay with NTM status. While there was no significant interaction between modulator treatment (binary) and NTM group on lung function ($F_{1,53}$ = 3.08, $P$ = 0.085), there were significant impacts on diversity and dominance (Fig. 2). The results indicated that even accounting for a significant effect of antibiotic treatment ($F_{1,54}$ = 8.11, $P$ = 0.006), the modulator treatment had a significant impact on microbiome diversity ($F_{1,54}$ = 9.99, $P$ = 0.003), with those on modulator therapy having a higher number of taxa present (mean Fisher's alpha = 1.93 ± 1.17) than those not on modulators (mean Fisher's alpha = 0.96 ± 0.48). This trend continued, even though there was no significant ($F_{1,54}$ = 3.43, $P$ = 0.070) effect of antibiotics, with those on modulators shown as having a significantly ($F_{1,54}$ = 12.94, $P$ = 0.001) less dominated (mean Berger-Parker index = 0.30 ± 0.16) microbiome than those not on modulator treatment (mean Berger-Parker index = 0.49 ± 0.18).

The analysis has already indicated that NTM culture status and modulator therapy significantly altered diversity. The analysis continued to evaluate if there were significant interactions between infection and treatment. While the overall model indicated a nonsignificant interaction ($F_{1,52}$ = 3.07, $P$ = 0.086), the post hoc analysis revealed that there were significant differences, depending on the combination of whether the pwCF were

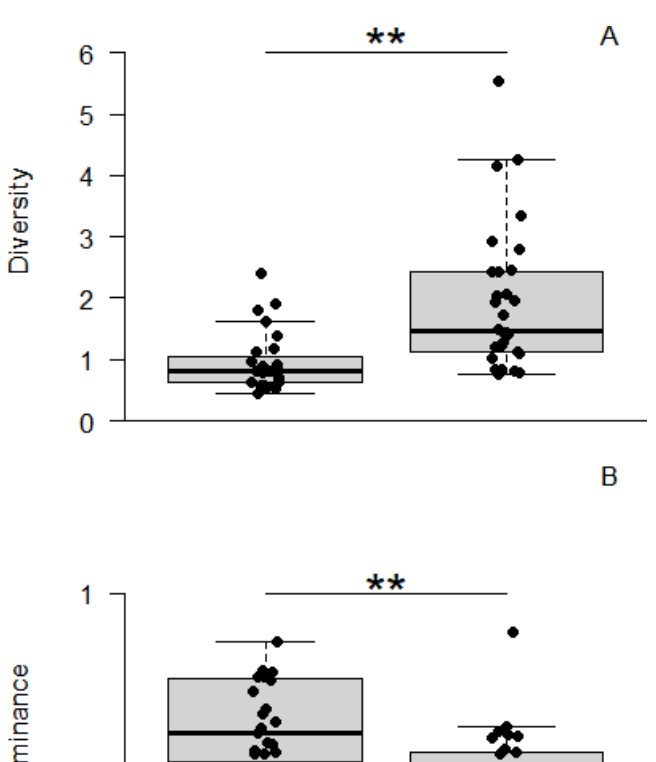

**FIG 2** The effect of modulators on microbiome diversity. Significant changes in microbiome diversity (Fisher's alpha) were observed for pwCF during modulator treatment, where diversity levels increase with modulators (A), with an accompanying decrease in how dominated (Berger-Parker) the communities were (B). The asterisk indicates a statistically significant Tukey's honestly significant difference result, where ** represents P values of <0.01.

undergoing CFTR modulator therapy and had a positive NTM culture. Here, NTM culture-negative pwCF with CFTR modulator therapy (mean Fisher's alpha = 2.24 ± 1.29) had a significantly ($P_{adj} < 0.046$) higher diversity than all the other groups (Fig. 3).

## Microbiome composition is remodeled by the presence of NTM

Given there was a significant influence of mycobacterial culture status on microbial alpha diversity, the analysis next focused on beta-diversity measures (Fig. 4). By analyzing community composition, the results indicated that there were significant differences (permutational multivariate analysis of variance [PERMANOVA] $F_{1,54} = 2.01$, $R^2 = 0.035$, $P = 0.007$) due to antibiotic treatment, and significant differences in community composition in those samples with different NTMs were detected (PERMANOVA $F_{3,54} = 1.32$, $R^2 = 0.07$, $P = 0.050$). Pairwise comparisons failed to find significant difference between paired comparisons ($P_{adj} > 0.302$).

These differences in composition were investigated and, after removing taxa that were significantly associated with antibiotic treatment ($n = 5$, Table S1), the analysis found that 10 species (Table S2) were significantly reduced in abundance in NTM culture-positive samples, including the canonical pathogen *Pseudomonas aeruginosa* ($P = 0.005$); anaerobic species *Prevotella histicola* ($P = 0.005$), *Veillonella nakazawae* ($P = 0.010$), and

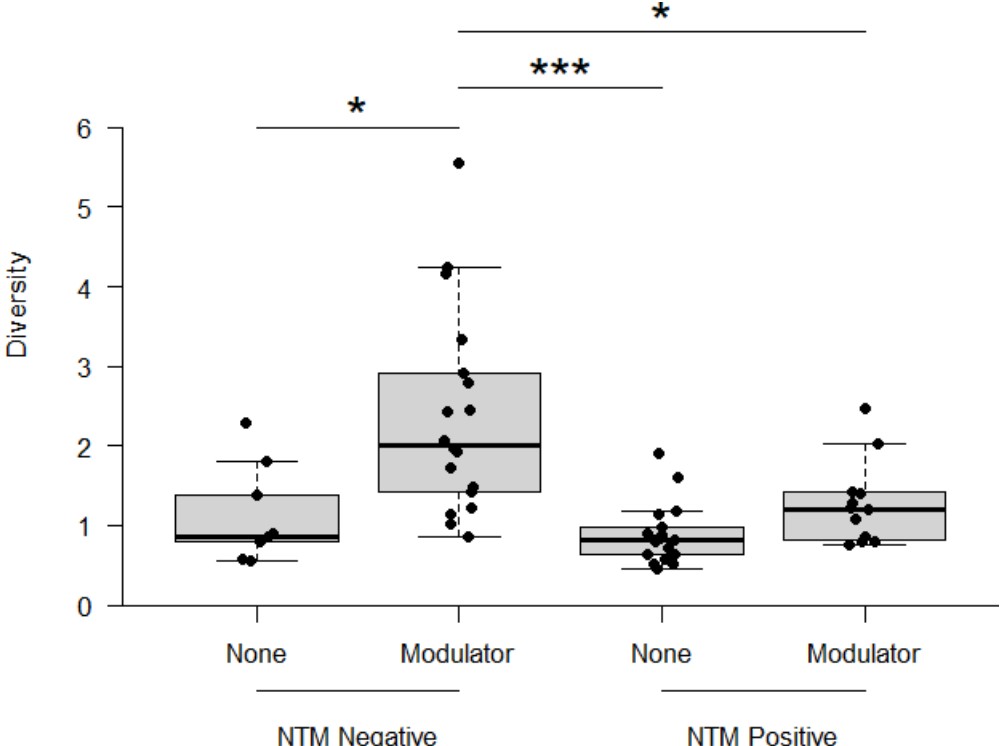

**FIG 3** Interactions of NTM culture status (positive/negative) and modulator therapy (modulator/none) on microbiome diversity. Samples from pwCF who were NTM culture negative and on CFTR modulator therapy had significantly more diverse microbiomes than any of the other combinations of NTM culture status and CFTR modulator therapy. The asterisk indicates a statistically significant Tukey's honestly significant difference result, where * and *** represent $P$ values of <0.05 and 0.001, respectively.

*Veillonella rogosae* ($P$ = 0.045); and commensal species *Streptococcus intermedius* ($P$ = 0.040) and *Gemella morbillorum* ($P$ = 0.015). Conversely, only one taxon significantly increased within the NTM-positive samples: *Achromobacter xylosoxidans* ($P$ = 0.035) .

## Modulator therapy enhances remodeling by NTMs

Finally, the impact of modulator therapy on microbiome composition was modeled together with the NTM group. This was to assess whether the taxonomy of NTM present had an interactive effect with modulator therapy on the microbiome. After accounting for the significant ($F_{1,48}$ = 2.08, $R^2$ = 0.04, $P$ = 0.005) influence of antibiotic treatment, there were clear significant ($F_{1,48}$ = 2.76, $R^2$ = 0.05, $P$ = 0.002) differences in microbiome composition attributed to the presence of a modulator (Fig. 5). There were also significant differences between the different NTM group present ($F_{3,48}$ = 1.37, $R^2$ = 0.07, $P$ = 0.029), suggesting that there are different consequences of CFTR modulator therapy with different *Mycobacterium* spp. present; however, no significant interaction between modulator therapy and the NTM group present was found ($F_{3,66}$ = 1.07, $R^2$ = 0.05, $P$ = 0.323).

Finally, the species were assessed for their association with modulator-NTM combinations. Overall, 17 species were found to have significant associations (Table S3); in particular, multiple *Prevotella histocola* ($P$ = 0.030) and *Streptococcus intermedius* ($P$ = 0.030) were significantly associated with pwCF groups on modulators or without NTM infection. In contrast, recognized pathogen *P. aeruginosa* was significantly associated ($P$ = 0.005) with both NTM-negative groups, regardless of CFTR modulator therapy, and

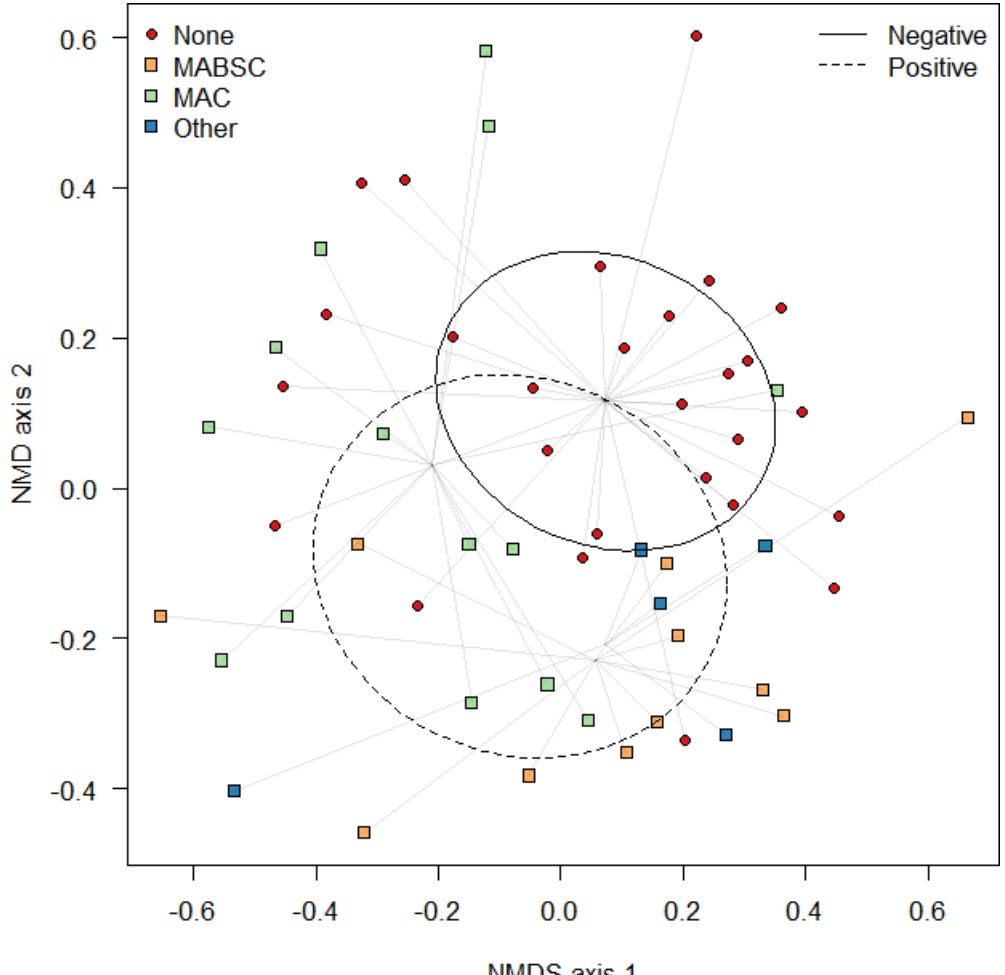

**FIG 4** Graphical representation of community dissimilarity in ordinal space using non-metric multidimensional scaling. Dissimilarities between the communities were measured using the Bray-Curtis dissimilarity index and plotted in ordinal space where points closest together are highly similar, whereas points far apart are highly dissimilar. Significant clusters were observed between samples that were NTM culture positive (dotted line, red circular points) and those that were negative (continuous line, square points), with subclusters (gray lines) indicating the NTM groups; MABSC (orange), MAC (green), and other (blue). Ellipses represent the standard deviation around the mean centroid for the cluster. Gray lines converge at the centroid for that cluster. NMDS, non-metric multidimensional scaling.

members of the *Burkholderia cepacia* complex (BCC, $P = 0.015$) and the genus *Staphylococcus* ($P = 0.015$) and *Haemophilus influenzae* ($P = 0.045$) were associated with groups without CFTR modulator therapy, regardless of NTM infection status.

## DISCUSSION

Previous research has shown that pwCF who have a microbiota with low diversity and high species dominance are associated with poorer clinical outcomes in relation to lung function (32, 46, 47). These individuals are also at higher risk of increased frequency of pulmonary exacerbation, which may lead to a faster progression of lung disease (47, 48). Infection with NTMs has also been associated with poor clinical outcomes and decline in pulmonary function (49, 50). Earlier research analyzing the community composition in CF lung microbiota and its relationship to NTM positivity is limited, therefore highlighting a research gap that needs to be addressed. While this study should be considered as an initial investigation, due to sample numbers included, predominantly due to the concurrent onset of the COVID-19 pandemic, and widespread uptake and use of

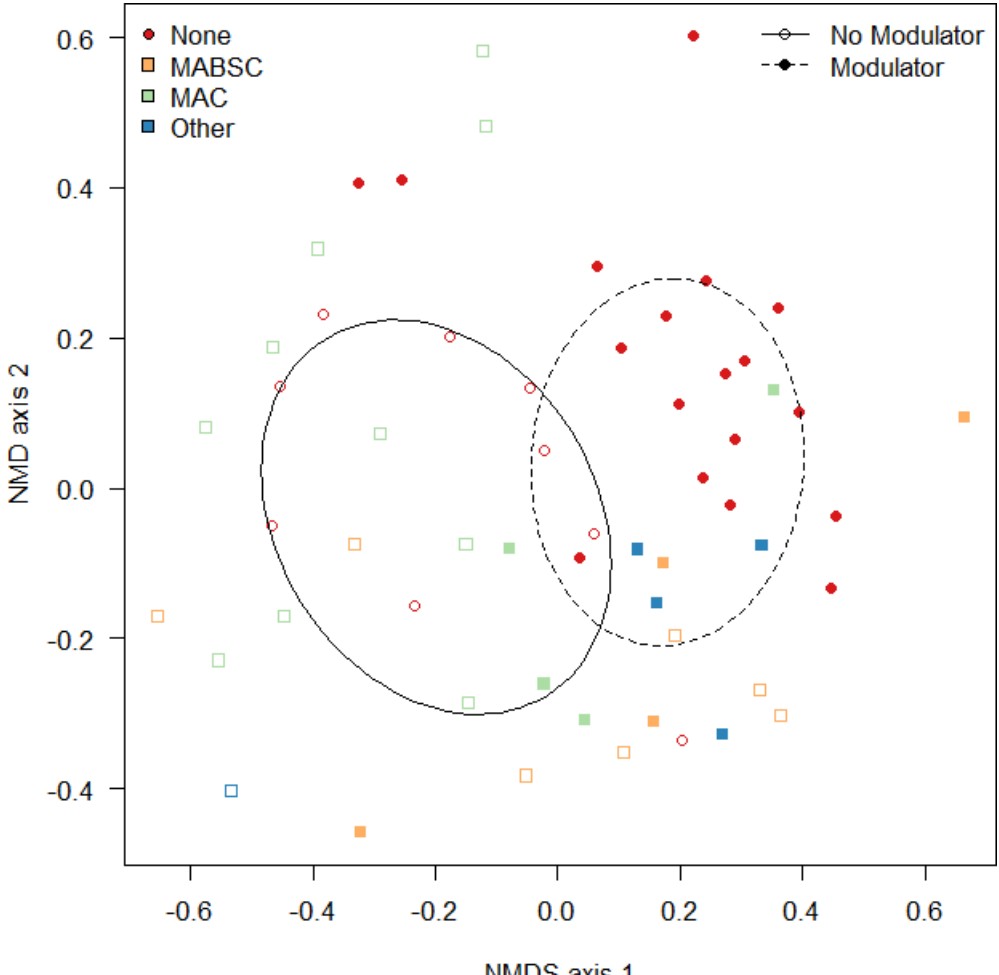

**FIG 5** Microbiome composition is significantly influenced by modulator therapy. Plotted in ordinal space using non-metric multidimensional scaling, there are significant differences between microbiomes with (filled points, dotted line) and without (open points, continuous line) CFTR modulator therapy. NTM groups are denoted as follows: MABSC (orange), MAC (green), and other (blue). Ellipses represent the standard deviation around the mean centroid for the cluster.

CFTR modulator therapies, occurring midway through recruitment, this study adopted a multifaceted approach to assess the overall microbiota, the type of NTM present in the microbiota, and the impact of CFTR modulator therapy. Our data highlight potential impacts that can be used to create hypotheses for future, larger studies.

Here, we present differences between the composition of the CF lung microbiota in NTM-positive and NTM-negative individuals and those on modulator therapy. In particular, the analysis indicated that there was a significant decrease in the frequency and abundance of *P. aeruginosa* in samples that were culture positive for NTM infection. Although the mechanisms underpinning this are unknown, this observation has been recorded previously (51), and interactions between species are increasingly shown to be key to lung functioning (34). We postulate that NTMs either interact antagonistically to some members of the microbiome or exploit vacant niches within the lung habitat, preventing subsequent colonization. This latter postulation could coincide with collateral effects of treatment with antibiotics, such as aminoglycosides (52), or the composition and niche occupation of the microbiome initially experienced by incoming pathogen (53), although further research is required. The analysis also indicated a relationship between modulator use and an increase in "commensal" bacterial species, as previously postulated (54). However, the high levels of BCC (55) members detected

in non-modulator-treated groups are concerning clinically . BCC is known to cause "cepacia syndrome," characterized by severe necrotizing pneumonia, respiratory failure, and bacteremia, with a high mortality rate and a contradiction to potentially lifesaving lung transplantation (56, 57); however, whether there is a mechanistic link between NTM and BCC prevalence is an area of future study.

Our analysis also provided evidence that while there were significant changes in the microbiome between those receiving CFTR modulator therapy and those not receiving it, NTM culture-positive pwCF remained compositionally distinct from NTM-negative samples, regardless of whether they were undertaking CFTR modulator therapy. This suggests that NTMs persist despite CFTR modulators, as with other pathogens (58), requiring the further study of the importance and pathogenesis of these organisms.

Our cohort included samples from individuals across the spectrum of pwCF; some were undergoing exacerbation, antibiotic treatment, or CFTR modulator therapy. As NTM infection, particularly chronic infection, affects between 2.6% (59) and 10.0% (60) of the pwCF population, it is vital to capture the widest remit possible to understand general aspects of NTM infection. This poses a dilemma: with the introduction of CFTR modulator therapies, it is no longer the norm that pwCF will produce sputum spontaneously, leading to the James Lind Alliance research priority in CF: "What is the best way to diagnose lung infection when there is no sputum?" (61). As such, to understand mechanisms of pathogenicity and the wider microbiome, it is necessary to have representation of the population regardless of sampling methodologies so as to not artificially bias the study. This does, however, bring further cofactors into the analysis, as shown here; sputum producers are more likely to be in exacerbation or not on modulators, with cough swab samples coming from those not attending a clinic. This begs the question of whether the microbiological discrepancies between sputum and cough swabs are truly greater than the natural variation between pwCF. Answering this question could be addressed by larger studies.

A further caveat is that the assignment of species taxonomy should be considered putative due to the length of the 16S rRNA gene sequence. A previous study (62) concluded that sequencing regions of 16S rRNA gene alone can be insufficient in distinguishing between closely related species, such as those from the BCC. Furthermore, due to the non-specific nature of 16S rRNA gene sequencing (63), NTM complexes are often underrepresented (64) and incorrectly identified (65, 66). Focused research in this area is needed to develop a high-throughput, culture-independent, method for identifying NTM alongside the wider microbiome.

In conclusion, the data and analysis presented here highlight potential effects of the presence of NTM and their influence on the respiratory microbiome, in particular, significant associations between NTM presence and decreasing *P. aeruginosa*. Due to the undefined consequences of NTM infection and clinical decline (9), it is not possible with this data set to attribute mechanistic causality; however, given the associations presented here, we believe this is clearly an area of clinical importance and future work.

## MATERIALS AND METHODS

### Participant recruitment

Adult pwCF were recruited from the University Southampton Hospital (UHS) NHS Trust, Cystic Fibrosis Center, UK, and The University of Vermont (UVM), USA (Table 1). PwCF who had no history of positive NTM culture were denoted as "NTM negative," and those who were culture positive and clinically defined as chronically colonized using the Leeds criteria (67) at the time of sampling were denoted as "NTM positive" and were subsequently subgrouped according to the species of NTM they were culture positive with MAC, MABSC, or other. Culture of pwCF respiratory samples was done and confirmed by UHS and UVM clinical pathology laboratories. Initially, participants were assessed, and respiratory samples were collected in the clinic during routine appointments by their

regular CF team. The collection method differed, with some samples being collected in the clinic (until March 2020) and others collected at home due to COVID-19 restrictions on clinic attendance and posted (from Sept 2020).

## Nucleic acid extraction

Prior to DNA extraction, sputum samples were centrifuged at 1,107 × *g* for 10 minutes at room temperature; the supernatant was discarded; and the pellet resuspended in 900 µL of phosphate-buffered saline (PBS). The process was repeated with the final pellet resuspended in 500 µL of PBS (30). To discriminate between live and dead cells, propidium monoazide (PMA) was used to covalently cross-link to DNA molecules, inhibiting amplification by PCR and thus excluding the dead/damaged cells from further analysis (68, 69). In brief, 500 µL of washed sputum was transferred into a 1.5 mL amber micro-centrifuge tube (Sigma-Aldrich, UK), and 1.25 µL of PMA (Biotium, USA) was added to each tube, then incubated for 15 minutes at room temperature. The mixture was then transferred into a clear micro-centrifuge tube then added to an LED lightbox for a further 15 minutes (70). PMA-sputum was added to a capped 1.5 mL microcentrifuge tube which was previously prepared with one tungsten carbide bead and glass beads (Merck, Dorset, UK) (70) and 400 µL of DNA/RNA lysis buffer (Zymo Research, USA). The samples were homogenized (FastPrep-24 Homogeniser; MP Biomedicals, Loughborough, UK) for two 30 second bursts. Nucleic acid extraction was performed following the manufacturer's instructions using Quick-DNA/RNA Miniprep Kit (Zymo Research). DNA was then stored at −20℃ for future use.

## Microbiome sequencing

The microbiome of the samples was assessed by two-step 16S rRNA gene amplicon-based sequencing using the Illumina MiSeq system. The first amplicon PCR was achieved using phased primer sets (71, 72) targeting the V4–V5 regions of the 16S rRNA gene (73–75). Each 25 µL PCR consisted of 12.5 µL 2× Q5 Master Mix (New England Biolabs, Hitchin, UK), 2 µL (0.2 µM) of phased primer pool (Invitrogen, Paisley, UK), and 8.5 µL of ultrapure, molecular-grade water (Cytiva, Fisher Scientific UK Ltd.) with the following parameters: 95℃ for 3 minutes, followed by 30 cycles at 95℃ for 30 seconds, 60℃ for 30 seconds and 72℃ for 30 seconds, ending with one cycle at 72℃ for 5 minutes. PCR reactions were confirmed by gel electrophoresis. Following successful amplicon amplification, PCR fragments of 350 bp and below were removed using AMpure XP PCR purification beads (Beckman Coulter Life Sciences, USA) following the manufacturer's instructions and quantified using a Qubit fluorometer (Thermo Fisher, UK). Illumina sequencing adapters were added to each sample in a second PCR. Each 25 µL PCR reaction consisted of 12.5 µL, 2× Q5 Master Mix (New England Biolabs), 1.25 mM of each primer, 5 µL cleaned PCR product, and 2.5 µL of ultrapure molecular grade water using the following parameters: 95℃ for 3 minutes, followed by 20 cycles at 95℃ for 30 seconds, 55℃ for 30 seconds and 72℃ for 30 seconds, ending with one cycle at 72℃ for 5 minutes. The fragment was confirmed using gel electrophoresis and then purified using AMpure beads, and fragment sizes were confirmed using the Agilent Bioanalyzer (Agilent, USA). The samples were sequenced on the Illumina MiSeq platform using the Illumina MiSeq V3 600 cycle reagent kit (Illumina Cambridge Ltd, Cambridge, UK).

## Microbiome sequence analysis

The analysis of raw sequence data was then performed through the DADA2 pipeline (39) using R (76) (v.4.3.2). Forward and reverse reads were truncated at 200 and 175 bp, respectively, with taxonomy assigned using the default matching parameters (100% identity) to the SILVA Database (v.138.1) (77). Unassigned amplicon sequence variants (ASVs) found were manually assigned using the Basic Local Alignment Search Tool (78) nucleotide database (79) and matched with sequences based on a minimum of 95% query coverage, with the lowest *e* value (34). Multiple sequences assigned to the same

ASV were condensed into OTUs for statistical analysis. Given the length of the ribosomal sequences analyzed, species identities should be considered putative.

## Statistical analysis

To assess the likelihood of explanatory variables co-occurring, binary regression was undertaken using a general linear model with binomial errors, generating the odd ratios and confidence intervals from the estimates. Clinical characteristics were modeled against lung function in a single analysis of variance with type III errors, which assessed the variation as if each variable was entered first in the model. To gauge the degree of change in the microbiome, each sample was examined for diversity (Fisher's alpha index of diversity), dominance (Berger-Parker index), and Bray-Curtis similarity measures. In addition to the similarity indices, non-metric multidimensional scaling was used to visualize the differences between groups. The significance of the alpha-diversity measures was determined using the Kruskal-Wallis analysis, and the beta diversity was tested for significance using Bray-Curtis-based PERMANOVA with 999 permutations. In all models, antibiotics (binary) were entered first into the model to account for variation associated with antibiotic treatment, as described in the text; all other variables were entered after, and their significance was calculated in order. Post hoc analyses were conducted using Tukey's honestly significant difference, and adjusted $P$ values ($P_{adj}$) were reported. Calculation of significant indicator species was undertaken using 100 permutations to assess whether individual species had higher (or lower) frequencies and abundances in one particular group compared to the others (80). All analysis and visualizations were conducted using R (76) (v.4.3.2) using the packages car (81), indicspecies (80), and vegan (82).

## ACKNOWLEDGMENTS

The authors thank the people with cystic fibrosis and staff at each of the contributing centers for their involvement, time, and patience in the sample collection.

## AUTHOR AFFILIATIONS

[1]Department of Life Sciences, Manchester Metropolitan University, Manchester, United Kingdom
[2]The Grange University Hospital, Cwmbran, Gwent, United Kingdom
[3]Department of Microbiology and Molecular Genetics, Larner College of Medicine, University of Vermont, Burlington, Vermont, USA
[4]Division of Pulmonary and Critic, Care Medicine, Department of Medicine, Larner College of Medicine, University of Vermont, Burlington, Vermont, USA
[5]Wessex Adult Cystic Fibrosis Unit, University Hospitals Southampton NHS Foundation Trust, Southampton, United Kingdom
[6]National Institute for Health Research, Southampton Biomedical Research Centre, Southampton, United Kingdom
[7]Department of Applied Sciences, Northumbria University, Newcastle, United Kingdom
[8]Department of Respiratory Medicine, Northern Care Alliance NHS Foundation Trust, Salford, United Kingdom
[9]Department of Natural Sciences, Manchester Metropolitan University, Manchester, United Kingdom

## PRESENT ADDRESS

Michelle Hardman, School of Biological Sciences, Faculty of Biology, Medicine and Health, University of Manchester, Manchester, United Kingdom

## AUTHOR ORCIDs

Christopher van der Gast  http://orcid.org/0000-0003-1101-4048

Damian W. Rivett ⓘ http://orcid.org/0000-0002-1852-6137

## FUNDING

| Funder | Grant(s) | Author(s) |
|---|---|---|
| Cystic Fibrosis Trust | VIA078 | Michelle Hardman |
| | | Thomas W. V. Daniels |
| | | Christopher van der Gast |
| | | Damian W. Rivett |
| Jean Shanks Foundation | | Sarah Higgi |

## AUTHOR CONTRIBUTIONS

Michelle Hardman, Data curation, Formal analysis, Investigation, Methodology, Validation, Writing – original draft | Sarah Higgi, Conceptualization, Resources | Liam Hanson, Investigation, Methodology | Kristin Schutz, Resources | Matthew J. Wargo, Resources, Writing – review and editing | Charlotte C. Teneback, Resources, Writing – review and editing | Thomas W. V. Daniels, Conceptualization, Funding acquisition, Resources, Supervision, Writing – review and editing | Christopher van der Gast, Conceptualization, Funding acquisition, Supervision, Writing – review and editing | Damian W. Rivett, Conceptualization, Formal analysis, Supervision, Writing – original draft

## DATA AVAILABILITY

The data sets generated and/or analyzed during the current study, along with the R script, are available on FigShare at 10.6084 /m9.figshare.26946850.v1 and in the Sequence Read Archive under BioProject number PRJNA1152493.

## ETHICS APPROVAL

Samples used in this study were collected from two cystic fibrosis clinics located in Southampton, UK, and Burlington, VT, USA. All samples were collected from pwCF, after obtaining written informed consent, under ethical approval from the centers involved. Samples from Southampton General Hospital were collected under approval from the NHS Research Ethics Committee (Ref: 06/Q1704/26). Samples from Burlington were collected under University of Vermont Institutional Review Board (CHRMS STUDY# M13-160).

This study was reviewed and approved by the National Research Ethics Committee (A), Southampton and southwest Hampshire, UK, and assigned the Research Ethics Committee reference number 08 /H0602/126 and by The University of Vermont Institutional Review Board in the USA under the Institutional Review Board number 00000485. People with cystic fibrosis provided written consent.

## ADDITIONAL FILES

The following material is available online.

### Supplemental Material

**Tables S1 to S3 (Spectrum00382-25-s0001.docx).** Species associations.

### Open Peer Review

**PEER REVIEW HISTORY (review-history.pdf).** An accounting of the reviewer comments and feedback.

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
