## [Reviewer comments · Microbiology Spectrum]

Microbiology Spectrum

Non-tuberculosis mycobacteria remodel lung microbiota in cystic fibrosis associated respiratory infections.

Michelle Hardman, Sarah Higgi, Liam Hanson, Kristin Schutz, Matthew Wargo, Charlotte Teneback, Thomas Daniels, Christopher van der Gast, and Damian Rivett

Corresponding Author(s): Damian Rivett, Manchester Metropolitan University

Review Timeline:

Submission Date:	February 13, 2025
Editorial Decision:	April 10, 2025
Revision Received:	June 5, 2025
Accepted:	June 29, 2025

Editor: Silvia Cardona

Reviewer(s): The reviewers have opted to remain anonymous.

Transaction Report:

DOI: <https://doi.org/10.1128/spectrum.00382-25>

Re: Spectrum00382-25 (Non-tuberculosis mycobacteria remodel lung microbiota in cystic fibrosis associated respiratory infections.)

Dear Dr. Damian W Rivett:

Thank you for submitting your manuscript to Microbiology Spectrum. Your work has been reviewed by two experts in the field. While the topic is considered important in the context of the microbial community of the cystic fibrosis lung and the impact of the new treatments, reviewers have voiced several concerns with the methodology and conclusions. As you know, the publication policy of Microbiology Spectrum focuses on accepting manuscripts that are technical and methodologically sound. If you feel you can address the reviewers' concerns, I am willing to receive a revised version of your manuscript. Please note that additional reviewers may evaluate the revised version.

Revision Guidelines

Sincerely,
Silvia Cardona
Editor
Microbiology Spectrum

Reviewer #1 (Comments for the Author)

While the microbiome in CF is an important topic, there are major methodological limitations and a lack of accounting for significant confounders in this manuscript, which reduce the validity of the conclusions. However, the authors can re-analyze the

data following the suggestions below to minimize possible confounding variables and then verify if their conclusions still stand solid and tone down the discussion

1. The data does not describe whether the patients were receiving antibiotic treatment. Please address and account for the antibiotic treatment. If necessary, regroup the patients according to this variable
2. Separately analyze the different specimen types. While cough swabs and sputum may have been analyzed before and significant differences may not have been found, it does not mean that this will be the case for the samples used in this work.
3. Revisit the clustering approach. The authors say they are generating ASVs and then collapsing into OTUs. However, this may not be the best approach. The authors should indicate what % similarity they are using for the individual features. Ideally, this would have 100% similarity ASVs.

Reviewer #2 (Comments for the Author):

In this manuscript, the authors report on results of 16S rRNA gene sequencing of a sputum and cough swabs sample from people with cystic fibrosis (CF), and analyze differences in the airway microbiome between people with and without NTM infection, and those with and without CFTR modulator use. They identify certain taxa that differ between these groups, and conclude that "the presence of NTM significantly alters the diversity and composition of the lung microbiota in pwCF". The interplay between NTM infection, CFTR modulators, and CF clinical status is an important topic. Unfortunately, significant methodological issues limit these results. And overall, the language throughout the manuscript inappropriately suggests identification of causal relationships, rather than associations (e.g., "In conclusion, the data and analysis presented here demonstrate that the presence of NTMs significantly alter the composition of the microbiome").

The first major issue is in the omission of important clinical data:

- How long had the subjects in the NTM positive group had NTM infection? Were there differences in NTM infection duration? Twenty-five to thirty percent of people with CF who have a positive culture for NTM will have spontaneous clearance of the NTM, without directed treatment, and it seems likely that the impact that NTM may have on the microbiome (and clinical course) would differ in the setting of a transient infection compared to chronic infection.
- Antibiotic treatment data is not included. Antibiotics are arguably the biggest drivers of changes in airway microbiota. For people with CF and NTM infection, this is relevant in 2 ways. First, an incident NTM infection typically triggers clinical work up to determine if NTM disease is present, which includes management of other CF pathogens, in order to determine the relative contribution of the NTM to the clinical picture. It's common for new NTM infection to be followed, for example, by aggressive anti-pseudomonal treatment. Second, if NTM disease is diagnosed, this is treated with months to years of multiple NTM-directed antibiotics. Any data on microbiome differences associated with NTM infection needs to account for (or at least acknowledge) the impact of antibiotics (both for NTM and for other CF pathogen) used in the setting of NTM infection. My confidence that any of the identified associations are directly due to NTM infection, rather than reflective of differences in antibiotic use, is low.

Sputum and cough swab samples are combined in the analyses. There are two main issues with this:

- First, while the authors do not provide these details, it's likely that the distribution of sputum vs cough swab samples differed between the comparator groups (people on modulators are more likely to have cough swab samples, and sputum production is decreased, and people who are sampled with cough swabs are less likely to have NTM detected, as NTM culture typically requires a sputum sample).
- Second, multiple publications indicate that the airway microbiome and CF pathogen detection in CF differs when measured with sputum vs cough swabs.

E.g.: Zemanick ET, Wagner BD, Robertson CE, Stevens MJ, Szefer SJ, Accurso FJ, Sagel SD, Harris JK. Assessment of airway microbiota and inflammation in cystic fibrosis using multiple sampling methods. *Ann Am Thorac Soc*. 2015 Feb;12(2):221-9. doi: 10.1513/AnnalsATS.201407-310OC. PMID: 25474078; PMCID: PMC4342834.

Ronchetti K, Tame JD, Paisley C, Thia LP, Doull I, Howe R, Mahenthalingam E, Forton JT. The CF-Sputum Induction Trial (CF-SpIT) to assess lower airway bacterial sampling in young children with cystic fibrosis: a prospective internally controlled interventional trial. *Lancet Respir Med*. 2018 Jun;6(6):461-471. doi: 10.1016/S2213-2600(18)30171-1. Epub 2018 May 16. PMID: 29778403; PMCID: PMC5971213.

The authors provide a reference, #57, to support this, but I don't see how this reference is relevant to analyzing sputum and cough swabs together.

Dear Editor

Please find below the responses to the reviewers' comments. The authors would like to thank both of them for their time spent on this and the constructive criticism. As such, we have taken the points raised on board and this prompted us to reevaluate some of the decisions made. There has, therefore, been several changes in the manuscript, detailed in the responses, to incorporate the reviewers' comments that have made this a more stringent and thought-provoking piece.

Major changes:

- 1) To fully incorporate the antibiotic data, we removed all samples (n=12) that did not have the required metadata. This has required a reanalysis, including antibiotics in the models.
- 2) The approach of mixed sampling has now been explicitly addressed in the manuscript, along with a new analysis of the issue, which hopefully addresses the concerns.

Once again, I, and the other authors, thank the reviewers for challenging our first submission and allowing us to make these improvements. Where we highlight edits, these are given as line numbers in the marked-up document.

Yours Faithfully,

Damian Rivett

Reviewer #1 (Comments for the Author)

While the microbiome in CF is an important topic, there are major methodological limitations and a lack of accounting for significant confounders in this manuscript, which reduce the validity of the conclusions. However, the authors can re-analyze the data following the suggestions below to minimize possible confounding variables and then verify if their conclusions still stand solid and tone down the discussion

We thank the reviewer for the time taken to review this manuscript and apologise for not including all the data to address these confounding factors. We have now included these analyses and addressed the tone of the discussion.

1. The data does not describe whether the patients were receiving antibiotic treatment. Please address and account for the antibiotic treatment. If necessary, regroup the patients according to this variable

We completely agree with the reviewer and have taken steps to address this omission. Going back to the metadata, we realised that some (n=12) samples were missing this data. To make a valid inclusion of the antibiotic data were excluded those that had no data, reducing out dataset to 57. Whilst we found no effect of antibiotic treatment on lung function, there were, expectedly, significant variations in the α and β -diversity of the samples. As such, we decided to analyse the explanatory variables taking the presence of antibiotics into account by including "antibiotics" as the first explanatory term in all models run thereby accounting for variance associated with antibiotic treatment. Further, we analysed the

species contributions with and without antibiotics and subsequently discounted all species with a significant change after antibiotic treatment if they (n=5) were found to be significant in any other treatment.

These changes have been implemented throughout the manuscript.

2. Separately analyze the different specimen types. While cough swabs and sputum may have been analyzed before and significant differences may not have been found, it does not mean that this will be the case for the samples used in this work.

We are in agreement that this methodological issue needs addressing, however, we do not believe that separating the two groups is the most appropriate solution in this scenario. When discussing this, we decided to investigate whether there were trends that appeared in the data with respect to the presence of swab vs sputum (notwithstanding the impact of COVID-19 related isolation). As such, by conducting binomial regression and generating odd ratios we found that the presence of sputum is directly related to whether the pwCF was in a period of exacerbation, or not on modulators. We realised, as the reviewer may be keenly aware of, that sputum production is not the norm for pwCF on modulators, and they are more likely to produce sputum during an exacerbation. As exacerbations trigger clinical treatment, they are also more likely to be administered antibiotics. As such, we believe that by accounting for antibiotic administration we can alleviate some of the methodological bias, whilst retaining the pwCF and the effect of NTM infection on the microbiome.

This is a significant issue facing CF microbiology research. Whilst many studies can mitigate this somewhat by increasing sample numbers, the study of NTM infection is limited by the population who has this, and are screened for NTMs.

We outline our thinking, and our new analysis in the new section of the results (lines 150-171) and in the discussion (lines 330-350).

3. Revisit the clustering approach. The authors say they are generating ASVs and then collapsing into OTUs. However, this may not be the best approach. The authors should indicate what % similarity they are using for the individual features. Ideally, this would have 100% similarity ASVs.

We have added the taxonomy assignment parameters into the methodology (line 445), which was indeed to 100% matching to an ASV. However, we stand by the collapsing on ASVs into OTUs due to the fragment size of the amplicons used even assignment at a species level is tentative and should be adopted with caveats (lines 364-371, 450). We are of the opinion that inferring biological or ecological meaning to ASVs above a putative species-level assignment, i.e. strain levels, at this length of a single phylogenetic gene may muddy an already busy manuscript.

Reviewer #2 (Comments for the Author):

In this manuscript, the authors report on results of 16S rRNA gene sequencing of a sputum and cough swabs sample from people with cystic fibrosis (CF), and analyze differences in the airway microbiome between people with and without NTM infection, and those with and without CFTR modulator use. They identify certain taxa that differ between these groups,

and conclude that "the presence of NTM significantly alters the diversity and composition of the lung microbiota in pwCF". The interplay between NTM infection, CFTR modulators, and CF clinical status is an important topic. Unfortunately, significant methodological issues limit these results. And overall, the language throughout the manuscript inappropriately suggests identification of causal relationships, rather than associations (e.g., "In conclusion, the data and analysis presented here demonstrate that the presence of NTMs significantly alter the composition of the microbiome").

We thank the reviewer for the time spent on this review and appreciate the in-depth comments provided. We address the more specific comments below, but we have also reassessed the discussion and toned down the definitive statements e.g. "In conclusion, the data and analysis presented here highlight potential effects of the presence of NTM and their influence on the respiratory microbiome" line 361

The first major issue is in the omission of important clinical data:

-How long had the subjects in the NTM positive group had NTM infection? Were there differences in NTM infection duration? Twenty-five to thirty percent of people with CF who have a positive culture for NTM will have spontaneous clearance of the NTM, without directed treatment, and it seems likely that the impact that NTM may have on the microbiome (and clinical course) would differ in the setting of a transient infection compared to chronic infection.

We thank the reviewer for pointing this omission out. All of the NTM+ patients recruited were done so on the criteria that they were clinically described as chronically colonised. This has been added to the manuscript (line 122, Table 1, 384)

-Antibiotic treatment data is not included. Antibiotics are arguably the biggest drivers of changes in airway microbiota. For people with CF and NTM infection, this is relevant in 2 ways. First, an incident NTM infection typically triggers clinical work up to determine if NTM disease is present, which includes management of other CF pathogens, in order to determine the relative contribution of the NTM to the clinical picture. It's common for new NTM infection to be followed, for example, by aggressive anti-pseudomonal treatment. Second, if NTM disease is diagnosed, this is treated with months to years of multiple NTM-directed antibiotics. Any data on microbiome differences associated with NTM infection needs to account for (or at least acknowledge) the impact of antibiotics (both for NTM and for other CF pathogen) used in the setting of NTM infection. My confidence that any of the identified associations are directly due to NTM infection, rather than reflective of differences in antibiotic use, is low.

We are in complete agreement that the antibiotic regimen should be taken into account. The reviewer rightly highlights that chronic infection leads to continuous clinical management through antibiotic administration. As such, after discussions we have returned to the original data and removed all those samples without appropriate data on antibiotic usage at the time of sampling (n=12). We have implemented a full re-analysis incorporating antibiotic treatment as the initial variable to account for variation associated with treatment. This analysis was conducted with antibiotic treatment entered as a binary variable as we believe this is the most appropriate way of accounting for these differences whilst maintaining generalisable conclusions (line 161-163)

Sputum and cough swab samples are combined in the analyses. There are two main issues with this:

-First, while the authors do not provide these details, it's likely that the distribution of sputum vs cough swab samples differed between the comparator groups (people on modulators are more likely to have cough swab samples, and sputum production is decreased, and people who are sampled with cough swabs are less likely to have NTM detected, as NTM culture typically requires a sputum sample).

-Second, multiple publications indicate that the airway microbiome and CF pathogen detection in CF differs when measured with sputum vs cough swabs.

E.g.: Zemanick ET, Wagner BD, Robertson CE, Stevens MJ, Szeffler SJ, Accurso FJ, Sagel SD, Harris JK. Assessment of airway microbiota and inflammation in cystic fibrosis using multiple sampling methods. *Ann Am Thorac Soc*. 2015 Feb;12(2):221-9. doi: 10.1513/AnnalsATS.201407-310OC. PMID: 25474078; PMCID: PMC4342834.

Ronchetti K, Tame JD, Paisley C, Thia LP, Doull I, Howe R, Mahenthalingam E, Forton JT. The CF-Sputum Induction Trial (CF-SpIT) to assess lower airway bacterial sampling in young children with cystic fibrosis: a prospective internally controlled interventional trial. *Lancet Respir Med*. 2018 Jun;6(6):461-471. doi: 10.1016/S2213-2600(18)30171-1. Epub 2018 May 16. PMID: 29778403; PMCID: PMC5971213.

The authors provide a reference, #57, to support this, but I don't see how this reference is relevant to analyzing sputum and cough swabs together.

Prompted by this comment, we re-evaluated our assumptions, and conducted a new analysis to understand how our samples were distributed. As we now present in the manuscript (lines 150-171) how the distribution of the swabs:sputum were in our samples. As such, we found a series of associations that demonstrated that the patient population was highly complex, and the presence of NTM did indeed associate with sputum production in most, but not all, cases. The upshot of this extra analysis, was that if we accounted for the antibiotic, then the confounding factors, except NTM positive culture, were accounted for so the resulting variation should down to the presence of NTMs.

We believe that this is a robust way of proceeding, however, this is an interesting point that has implication for many studies in the CFTR modulator therapy era and how to address the aspect of mixed sampling whilst maintaining an unbiased sample set. We have added this to our discussion (lines 330-350)

Re: Spectrum00382-25R1 (Non-tuberculosis mycobacteria remodel lung microbiota in cystic fibrosis associated respiratory infections.)

Dear Dr. Damian W Rivett:

Your manuscript has been accepted, and I am forwarding it to the ASM production staff for publication. Your paper will first be checked to make sure all elements meet the technical requirements. ASM staff will contact you if anything needs to be revised before copyediting and production can begin.

There is one spelling error in the title, and in the manuscript, "Nontuberculous" instead of "Non-tuberculosis". Please ensure the ASM production staff notices and corrects in the proofs.

Otherwise, you will be notified when your proofs are ready to be viewed.

Sincerely,
Silvia Cardona
Editor
Microbiology Spectrum

Reviewer #2 (Comments for the Author):

The authors have responded well to the prior comments, and have adequately addressed prior concerns regarding antibiotic and sampling variables.

My one minor comment is to please address the spelling error in the title, and in the manuscript, and use "Nontuberculous" instead of "Non-tuberculosis" throughout.